# Effects of AMF on Maize Yield and Soil Microbial Community in Sandy and Saline Soils

**DOI:** 10.3390/plants13152056

**Published:** 2024-07-25

**Authors:** Li Fan, Peng Zhang, Fuzhong Cao, Xueping Liu, Minjia Ji, Min Xie

**Affiliations:** 1College of Horticulture and Plant Protection, Inner Mongolia Agricultural University, Huhhot 010018, China; fanli@imau.edu.cn; 2College of Agriculture, Inner Mongolia Agricultural University, Hohhot 010019, China; 3Erdos Agricultural and Animal Husbandry Technology Extension Center, Erdos 017200, China; ordosnmjs_xccy@163.com (F.C.); jiminjia1991@126.com (M.J.)

**Keywords:** arbuscular mycorrhizal fungi, maize, yield, microbial community, sustainable agriculture

## Abstract

This study aimed to investigate the effects of applying arbuscular mycorrhizal fungi (AMF) on maize root growth and yield formation under different soil conditions. This study was conducted under sandy soil (S) and saline–alkali soil (Y), with treatments of AMF application (AM) and no AMF application (CK). The root characteristics, yield, and quality of maize were measured. High-throughput sequencing technology was employed to assess the impact of AMF on the soil microbial community structure, and the correlation between soil microbes and soil physicochemical properties was elucidated. The results show that under both sandy and saline–alkali soil conditions, AMF application significantly enhanced maize root growth, yield, grain quality, and soil available nitrogen (AN), available phosphorus (AP), and available potassium (AK) contents compared to the CK treatment. Soil microbial Alpha diversity analysis indicated that AMF application effectively increased soil microbial diversity and richness. Principal coordinate analysis (PCoA) and microbial community structure analysis revealed significant differences in bacterial communities between AM treatment in sandy soil (SAM) and CK in sandy soil (SCK), and significant differences in both bacterial and fungal communities between AM treatment in saline–alkali soil (YAM) and CK in saline–alkali soil (YCK). Furthermore, significant correlations between microbial communities and soil physicochemical properties were found, such as AN, AP, AK, soil salinity (SS), and organic matter (OM) content. AMF application had a greater impact on bacterial communities than on fungal communities. This study demonstrated that the use of AMF as a bio-fungal fertilizer was effective in improving spring maize yields, especially in terms of yield increase and quality stability in sandy and saline soils, thereby contributing to safe and sustainable cropping practices.

## 1. Introduction

The desertification and salinization of arable land are significant factors that inhibit crop growth and reduce yields, causing substantial annual losses in production and profits [1]. Moreover, the improper use of chemical fertilizers, soil erosion, and soil pollution all contribute to the disruption of soil structure, further exacerbating desertification and salinization [2]. However, introducing naturally occurring beneficial microorganisms into the soil can improve soil structure, thereby mitigating the adverse effects of poor soil conditions on crop production [3]. Among these beneficial microorganisms is arbuscular mycorrhizal fungi (AMF), a type of endophytic fungi belonging to the phylum *Glomeromycota*, which forms symbiotic relationships with over 80% of terrestrial plants [4,5]. AMF effectively promotes host plant growth, enhances plant stress resistance, and maintains the stability of soil ecosystems [6]. Additionally, AMF can increase the surface area of the host plant’s roots, thereby influencing plant growth, development, and biomass accumulation, which in turn boosts maize yields [7]. Furthermore, AMF has a restorative effect on soil ecology and has shown great success in plant disease control, indicating promising application prospects [8].

The structure of soil microbial communities directly impacts plant growth and development [9]. AMF can interact with various soil microorganisms, such as rhizosphere bacteria like *Bacillus* and *Pseudomonas*, and soil fungi including *Aspergillus* and *Trichoderma*, leading to significant changes in the composition, diversity, and functionality of the soil microbial community [10]. The influence of AMF on soil microbial community structure can be achieved through various mechanisms. For example, AMF affected the physical and chemical properties of the soil by secreting enzymes, organic acids, and other substances, creating a more suitable growth environment for other microorganisms [11]. Noceto et al. found that the metabolic products produced by AMF could directly influence the growth and reproduction of other microorganisms in their study on the interaction between AMF and rhizosphere bacteria in crop production [12]. Additionally, Xu et al. conducted experiments by inoculating AMF in soils with different phosphorus levels and found that AMF could influence the composition and diversity of soil microbial communities through competition, symbiosis, and other relationships [13]. Overall, the interaction between AMF and soil bacteria and fungi forms a complex soil microbial network, which positively affects the stability of soil ecosystems [14].

In summary, this study compared the effects of applying AMF under different soil conditions on maize root architecture, yield formation, grain quality, soil physical and chemical properties, and microbial community structure. The aim was to investigate the mechanisms through which AMF influences maize growth and soil microbial communities, thereby providing a theoretical basis for the use of AMF as a biofertilizer to ensure increased and stable maize production.

## 2. Results

### 2.1. The Effects of AMF on Corn Root Growth and Yield Formation

#### 2.1.1. Impact of AMF on Maize Root Growth

Based on Table 1, it was found that the application of AMF significantly promoted root growth, with the most pronounced effects observed in saline–alkali soils. Specifically, SAM (AM treatment in sandy soil) showed increases of 44.15% in total root length, 23.25% in root surface area, and 4.92% in root volume compared to SCK (CK in sandy soil). Similarly, YAM (AM treatment in saline–alkali soil) exhibited increases of 112.53% in total root length, 111.42% in root surface area, and 106.99% in root volume compared to YCK (CK in saline–alkali soil). Overall, under the YAM treatment, maize roots were the most developed, measuring 9272.27 cm for total root length, 0.80 mm for average root diameter, 2335.28 cm^2^ for root surface area, and 48.27 cm^3^ for root volume.

#### 2.1.2. Impact of AMF on Maize Yield Formation

As shown in Table 2, the application of AMF significantly increased the effective number of ears and the number of grains per ear, thereby effectively enhancing the corn yield. The corn yield of SAM was 811.5 kg/666.67 m^2^ (1 mu = 666.67 square meters), representing increases of 25.77%, 7.56%, and 10.56% compared to SCK in terms of yield, effective ear number, and grain number per ear, respectively. The corn yield of YAM was 681.4 kg/666.67 m^2^, with increases of 18.63%, 3.72%, and 11.22% compared to YCK in terms of yield, effective ear number, and grain number per ear, respectively.

#### 2.1.3. Impact of AMF on Maize Grain Quality

Based on Table 3, the application of AMF increased the contents of crude protein, crude starch, and crude fat to varying degrees, effectively enhancing the quality of maize grains. Specifically, SAM had crude protein, crude starch, and crude fat contents of 95.16 mg/g, 593.32 mg/g, and 4.00%, respectively, which were increased by 12.10%, 4.13%, and 14.29% compared to those of SCK. YAM had crude protein, crude starch, and crude fat contents of 89.97 mg/g, 587.44 mg/g, and 4.20%, respectively, which were increased by 2.99%, 3.50%, and 13.51% compared to those of YCK.

### 2.2. Effects of AMF on Rhizosphere Soil Physicochemical Properties and Microbial Community Structure

#### 2.2.1. Impact of AMF on Soil Physicochemical Properties

As shown in Table 4, under different soil conditions, the application of AMF led to an increasing trend in the ammonium nitrogen, available phosphorus, available potassium, and organic matter contents of the rhizosphere soils compared with CK. Specifically, the N, P, and K contents of SAM increased by 62.68%, 44.32%, and 158.55%, respectively, and the organic matter content increased by 166.67% compared with the SCK treatment. Additionally, the N, P, and K contents of YAM increased by 9.99%, 27.74%, and 67.58%, respectively, and the organic matter content increased by 55.56% compared with the YCK treatment.

#### 2.2.2. Impact of AMF on Soil Microbial Alpha Diversity

As shown in Figure 1, the Chao index, Shannon index, and Simpson index of bacteria and fungi were determined to explore the effect of AMF on the Alpha diversity of soil microorganisms under different soil conditions. Figure 1A–C show the diversity changes in rhizosphere soil bacterial communities. Under saline conditions, the application of AMF effectively increased the abundance and diversity of rhizosphere bacteria in maize. The Chao index and Shannon index of YAM were significantly increased by 17.12% and 2.87%, respectively, compared with those for YCK, while the Simpson index of YAM was significantly decreased by 17.24% compared with that for YCK. Figure 1D–F show the changes in the diversity of rhizosphere soil fungal communities under sandy soil conditions. The application of AMF effectively increased the abundance and diversity of rhizosphere soil fungi in maize, as shown by significant increases of 25.20% and 14.51% in the Chao index and Shannon index of SAM compared with SCK, and a significant decrease of 139.91% in the Simpson index. In contrast, under saline conditions, the Chao index and Shannon index of YAM were significantly increased by 222.70% and 19.90%, respectively, while the Simpson index of YAM was significantly decreased by 63.20% compared with that of YCK.

#### 2.2.3. Impact of AMF on Beta Diversity of Soil Microorganisms

As shown in Figure 2A, the PC1 and PC2 axes explained 56.76% and 24.69% of the differences in the bacterial community composition of maize rhizosphere soils, respectively, with a total contribution value of 81.45%. Among them, the differences between bacterial communities in sandy soils and saline soils were obvious on the PC1 axis, indicating that soil type influenced the changes in the bacterial community composition of maize rhizosphere soils. On the PC2 axis, the difference between bacterial communities with AMF application and CK under the same soil type was obvious, indicating that the application of AMF could effectively change the bacterial community structure of maize rhizosphere soil under the same soil condition. In addition, as shown in Figure 2B, PC1 and PC2 explained 34.66% and 18.12% of the differences in the fungal community composition of maize rhizosphere soil, respectively, with a total contribution value of 52.78%. Among them, the difference between YAM and YCK fungal communities was obvious on the PC2 axis, indicating that under saline and alkaline conditions, the application of AMF could effectively change the fungal community structure of maize rhizosphere soil.

#### 2.2.4. Impact of AMF on Soil Microbial Community Structure

As shown in Table 5, at the bacterial level, a total of 42 phyla, 134 classes, 331 orders, 545 families, 997 genera, and 1769 species were detected in the maize rhizosphere soil. At the fungal level, a total of 8 phyla, 25 classes, 59 orders, 119 families, 201 genera, and 271 species were detected in the maize rhizosphere soil.

As shown in Figure 3A, the top 10 taxa in relative abundance at the bacterial phylum level included *Proteobacteria*, *Chloroflexi*, *Actinobacteriota*, *Acidobacteriota*, *Gemmatimonadota*, *Bacteroidota*, *Myxococcota*, *Firmicutes*, *Patescibacteria*, and *Desulfobacterota*. Among them, *Proteobacteria* (22.70~34.63%), *Chloroflexi* (10.76~16.66%), *Actinobacteriota* (9.77~21.74%), *Acidobacteriota* (7.08~16.24%), *Gemmatimonadota* (4.76~8.52%), and *Bacteroidota* (5.75~6.51%) were the dominant phyla, with a total relative abundance of 75.41~83.15%. Moreover, at the fungal phylum level (Figure 3B), *Ascomycota* (43.83~64.55%), *Fungi_unclassified* (15.91~48.22%), *Basidiomycota* (3.36~13.50%), and *Zygomycota* (3.15~7.72%) were the dominant phyla, with a total relative abundance of 97.93~99.27%.

#### 2.2.5. Correlation Analysis between Soil Microbial Community and Soil Environmental Factors

As shown in Figure 4, the relationship between soil factors and microbial phyla was assessed using Pearson correlation. Under sandy soil conditions, *Proteobacteria* and *Desulfobacterota* in the bacterial phylum showed a significant negative correlation with AN, AP, AK, SS, and OM in soil, whereas *Chloroflexi*, *Actinobacteriota*, and *Acidobacteriota* exhibited a significant positive correlation with AP, AK, SS, and OM in soil (Figure 4A). In the fungal phylum, *Basidiomycota* and *Zygomycota* showed a significant positive correlation with AP, AK, SS, and OM in soil (Figure 4B). Additionally, under saline conditions, *Gemmatimonadota* and *Patescibacteria* in the bacterial phylum were significantly negatively correlated with AP, AK, SS, and OM in soil (Figure 4C); *Ascomycota* in the fungal phylum exhibited a significant negative correlation with AP, AK, SS, and OM in soil, whereas *Fungi_unclassified* and *Zygomycota* showed a significant positive correlation with AN, AK, and OM in soil (Figure 4D).

## 3. Discussion

The application of AMF effectively promoted plant root growth, thereby enhancing the plant’s ability to access water and nutrients [15]. Liu et al. found that the application of AMF significantly increased the plant root volume, number of lateral roots, and root length, thereby enhancing the adaptability of plant roots and promoting nutrient absorption [16]. This study yielded similar results, showing significant increases in maize root length, root surface area, and root volume under both sandy and saline–alkali soil conditions compared to the CK treatment after AMF application. Chen et al. found that the average diameter of *citrus* roots decreased after inoculation [17]. Similar results were obtained in this experiment, where under saline conditions, there was no significant change in the maize root average diameter between AMF and CK treatments. However, under sandy soil conditions, the average diameter of maize roots decreased after AMF application, which may be attributed to the promotion of lateral root formation after inoculation, leading to an increase in lateral root number and thus a decrease in average diameter [18,19].

AMF application as a biofertilizer was an important approach to improving maize yield and quality. Li et al. found, through their study on the relationship between crop agronomic traits, root systems, and microbes after inoculating with AMF, that the increase in crop yield is attributed to the enhanced microbial abundance, which promotes nutrient absorption by roots, thereby increasing crop yield [20,21]. The results of this study indicate that maize yield significantly increases after AMF application in both sandy and saline–alkali soils. Maize yield increased by 25.77% and 18.63% in sandy and saline–alkali soils, respectively. The significant increase in effective ears and grain number was a direct factor contributing to the increased maize yield [22]. The fundamental factor might be the increase in soil microbial diversity and abundance after AMF application, which promoted maize root growth, enhanced nutrient acquisition ability, and consequently increased maize yield [23]. Regarding grain quality, N. Massa et al. found that under reduced fertilization conditions, sole AMF application increased grain protein content, while co-application with rhizobia increased grain starch content [24]. This study yielded similar results, suggesting that AMF application could enhance maize nitrogen uptake [25], thereby effectively increasing grain protein, starch, and fat content.

AMF, soil, and soil microorganisms interact closely [26,27]. Wei et al. found that the application of AMF altered the soil physicochemical properties, thereby influencing the structure of soil microbial communities. Moreover, it could directly impact changes in the structure of soil microbial communities [28]. The results of this study indicate that in sandy soil, AMF application significantly increased fungal diversity and abundance. In saline–alkali soils, AMF application significantly enhanced the diversity and abundance of both bacteria and fungi. This might be due to the provision of more active substrates to the soil after AMF inoculation [29], and interactions between introduced AMF and other microbes [30], thereby increasing microbial community diversity and abundance. Additionally, PCoA in this study revealed that different soil types were the main factors causing variations in soil microbial community structure [31]. Furthermore, under the same soil conditions, AMF application led to differences in soil microbial community structure, with a greater impact observed on bacterial communities compared to fungal communities.

Different soil conditions affect the structure of soil microbial communities. Favorable soil environments may positively influence soil microbes [32]. In addition, microorganisms also respond differently to perceive different soil properties, thus affecting soil fertility [33]. In this study, under sandy soil conditions, bacterial communities showed a stronger correlation with soil physicochemical properties compared to fungal communities. Conversely, under saline–alkali soil conditions, fungal communities exhibited a stronger correlation with soil physicochemical properties than bacterial communities. This could be attributed to the direct or indirect effects of AMF application, which can alter microbial community structures and reach new equilibria [34]. However, the influence of AMF on soil microbes varied diversely across different soil conditions, sometimes promoting, having no effect, or even inhibiting microbial growth [35]. Therefore, microbial communities demonstrate varied correlations with soil physicochemical properties across different soil conditions.

## 4. Materials and Methods

### 4.1. Experimental Site Overview and Experimental Materials

The experiment was conducted from April to October 2023 in Xishe Village, Wang’aizhao Town, Dalad Banner, Ordos. The experimental site has a temperate continental monsoon climate. The soil types included sandy loam and saline–alkali soil. The basic soil properties of the sandy loam soil were pH 8.24, ammonium nitrogen content of 1.23 mg/kg, available phosphorus content of 7.30 mg/kg, and available potassium content of 87.55 mg/kg. The basic soil properties of saline–alkali soil were pH 8.10, ammonium nitrogen content of 1.23 mg/kg, available phosphorus content of 2.24 mg/kg, and available potassium content of 34.94 mg/kg. Additionally, the tested maize variety was Kehe 699. The arbuscular mycorrhizal fungi were obtained from the Rhizosphere Biology Research Institute of Yangtze University. The initial host of the AMF (*Funneliformis mosseae*) inoculant used in this study was saffron, produced in Dangxiong, Tibet. It was subsequently expanded with white clover and corn. The inoculum was a mixture of rhizomes and soil with a spore count of 13 spores/gram. The inoculum itself does not contain any bacteria.

### 4.2. Experimental Design

In this study, AMF application trials were set up for different soil types. Four treatments were established with three replications for each treatment: AMF applied in sandy soil (SAM), no AMF applied in sandy soil (SCK), AMF applied in saline soil (YAM), and no AMF applied in saline soil (YCK). Among them, the SAM and YAM treatments received 1 kg of fungal fertilizer per 666.67 m^2^ at the corn pulling stage, and each plot area was 45 m^2^ (4.5 m × 10 m). Other management measures, such as irrigation, weed control, and pest management, were kept consistent with standard field management practices to ensure uniformity across all treatments.

### 4.3. Sample Collection and Analysis

#### 4.3.1. Determination of Soil Physicochemical Parameters

Before planting and at the maturity of the maize, soil samples were collected from a 10 to 30 cm depth, with surface soil removed prior to sampling. Three random points were sampled per plot as composite samples. Soil samples were air-dried and passed through a 2 mm sieve to remove non-soil materials. Soil pH was measured using a pH meter (FE28-TRIS, Manufactured by Mettler Toledo Instruments (Shanghai) Ltd. in Xuhui District, Shanghai, China). Soil ammonium nitrogen (AN), available phosphorus (AP), and available potassium (AK) were determined using a soil nutrient rapid analyzer (TPY-8A, Manufactured by Topunon Ltd. and located in China (Hangzhou) Intelligent Information Industry Park). Soil organic matter (OM) was measured using the potassium dichromate oxidation method followed by titration with ferrous ammonium sulfate.

#### 4.3.2. Soil Microbial Community Analysis

Soil samples collected at maturity underwent DNA extraction, and the extracted genomic DNA was verified by 1% agarose gel electrophoresis. AMF-specific primers were chosen for the nested PCR amplification of the extracted DNA. The bacterial 16S rRNA V3-V4 region was amplified using primers 338F (5′-ACTCCTACGGGGAGGCAGCAG-3′) and 806R (5′-GGACTACHVGGGGIWTCTAAT-3′), while the fungal ITS2 region was amplified using primers ITS1F (5′-CTTGGTCATTTAGAGGAAGTAA-3′) and ITS2R (5′-GCTGCGTTCTTCATCGATGC-3′). TransGen AP221-02: TransStart Fastpfu DNA Polymerase was utilized. PCR products from the same samples were pooled and analyzed via 2% agarose gel electrophoresis. The PCR products were then recovered by excising the gel using the AxyPrepDNA Gel Recovery Kit (AXYGEN Corp., Union City, CA, USA), eluted with Tris_HCl, and assessed by 2% agarose gel electrophoresis. Purified PCR products were subsequently submitted to Yuanxin Biological Co., Ltd. (Zizhu Hi-Tech Park, Minhang, Shanghai, China). for paired-end sequencing using the Illumina PE250 platform (San Diego, CA, USA).

#### 4.3.3. Measurement of Rhizosphere Configuration

Three maize roots were collected from each plot at maturity, and root morphological parameters, including total length, surface area, diameter, and volume, were determined using an Epson Scan scanner (Epson 10000xl Root Scanner, manufactured by Epson (China) Limited, located in Dongguan City, Guangdong Province, China).

#### 4.3.4. Determination of Maize Yield

Corn from each plot was harvested at maturity and used to determine yield, thousand kernel weight, cob weight, and individual cob weight, and corn kernel yield was calculated based on 14% moisture content.

#### 4.3.5. Determination of Maize Grain Quality

Corn kernel crude protein, crude starch, and crude fat were determined using a FOSS near-infrared grain analyzer (the FOSS Infratec TM 1241 near-infrared grain quality analyzer is manufactured by Foss (Beijing) Science and Trade Co., Ltd., Beijing, China).

### 4.4. Data Analysis

Microsoft Excel 2019 was used for data processing. One-way analysis of variance (one-way ANOVA) was performed using SPSS 26 software, and editable plots were drawn using Origin 2022 software. The computer was equipped with a GeForce GTX 1650 graphics card with 6 GB video memory and an Intel(R) Core (TM) i7-9750H processor operating at 2.59 GHz.

## 5. Conclusions

This study analyzed the effects of AMF application on maize root growth, yield, and grain quality under different soil conditions, and further examined changes in soil microbial community structure. The potential value of AMF in maize production was evaluated in terms of root growth, yield formation, and microbial structure. Under two different soil conditions, AMF application effectively promoted maize root growth, significantly increased maize yield, and improved grain quality. Additionally, AMF application enhanced the diversity and abundance of the rhizosphere soil microbial community, thereby improving the soil’s physicochemical properties. Overall, AMF application created a richer soil microenvironment under different soil conditions, leading to increased maize production. Therefore, using AMF as a biofertilizer in maize production holds great potential for application in sandy or saline–alkaline soils.

## Figures and Tables

**Figure 1 plants-13-02056-f001:**
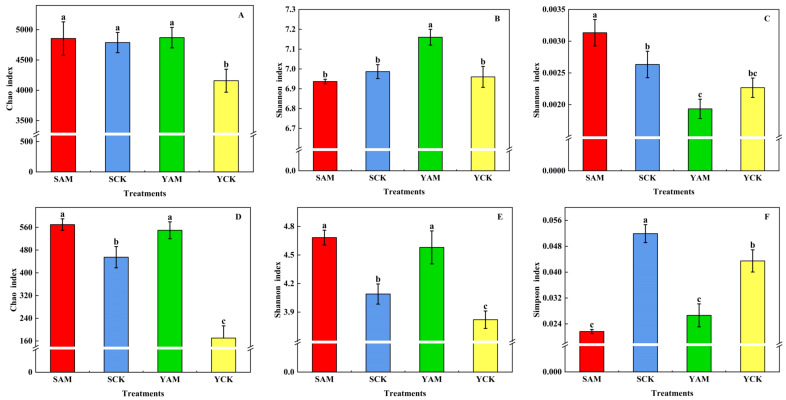
Effect of AMF on bacterial (**A**–**C**) ((**A**): Chao index of bacterial, (**B**): Shannon index of bacterial, (**C**): Simpson index of bacterial) and fungal (**D**–**F**) ((**D**): Chao index of fungal, (**E**): Shannon index of fungal, (**F**): Simpson index of fungal) Alpha diversity in soil. (Different letters in the figure indicate significant differences (*p* < 0.05)).

**Figure 2 plants-13-02056-f002:**
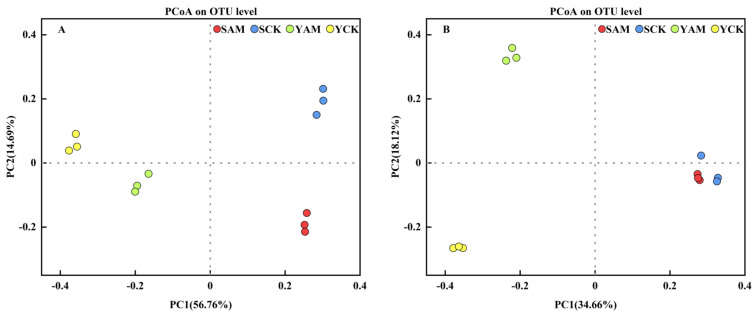
Principal coordinate analysis (PCoA) of bacterial (**A**) and fungal (**B**) communities in soil.

**Figure 3 plants-13-02056-f003:**
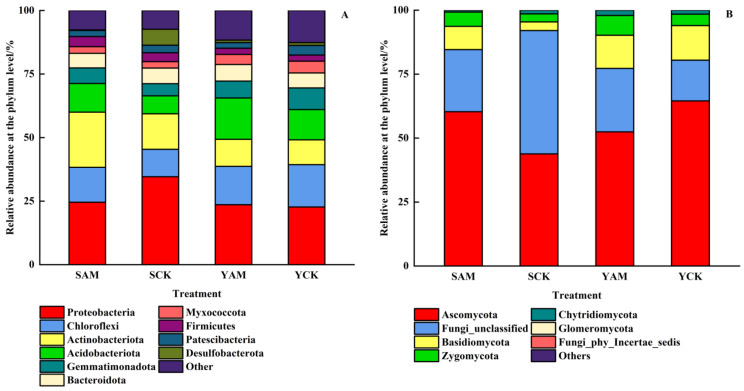
Bacterial (**A**) and fungal (**B**) phylum level community composition in soil.

**Figure 4 plants-13-02056-f004:**
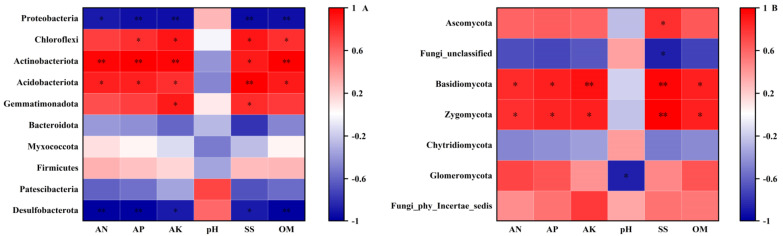
Correlation of bacterial (**A**,**C**) and fungal (**B**,**D**) communities in soil with soil physical and chemical properties. ** represent extremely significance at *p* ≤ 0.01 level; * represent significance at *p* ≤ 0.05 level.

**Table 1 plants-13-02056-t001:** Effect of AMF on root growth of maize.

Treatment	Total Length (cm)	Average Diameter (mm)	Surface Area (cm^2^)	Volume (cm^3^)
SAM	8909.45 a	0.76 b	2110.03 b	40.71 b
SCK	6180.89 b	0.87 a	1712.03 c	38.80 b
YAM	9272.27 a	0.80 b	2335.28 a	48.27 a
YCK	4362.90 c	0.80 b	1104.59 d	23.32 c

Results followed by different letters were significantly different according to Tukey’s post hoc test following ANOVA (*p* < 0.05). Same as below.

**Table 2 plants-13-02056-t002:** Effect of AMF on yield and its components.

Treatment	Yield (kg/mu)	Effective Panicles Number (Number/mu)	Grain Number per Spike (Grain)	Hundred-Grain Weight (g)
SAM	811.5 a	5063 a	534 b	35.5 a
SCK	645.2 c	4707 b	483 c	33.4 ab
YAM	681.4 b	4658 b	545 a	32.0 bc
YCK	574.4 d	4491 c	490 c	30.8 c

1 mu = 666.67 square meters.

**Table 3 plants-13-02056-t003:** Effect of AMF on maize kernel quality.

Treatment	Crude Protein (mg/g)	Crude Starch (mg/g)	Crude Fat (%)
SAM	95.16 b	593.32 a	4.00 ab
SCK	84.89 a	569.81 b	3.50 c
YAM	89.97 ab	587.44 b	4.20 a
YCK	87.36 b	567.56 ab	3.70 bc

**Table 4 plants-13-02056-t004:** Effect of AMF on soil physicochemical properties.

Treatment	AN (mg/kg)	AP (mg/kg)	AK (mg/kg)	pH	SS (%)	OM (%)
SAM	6.48 a	5.08 a	100.79 b	8.29 bc	0.25 c	1.60 a
SCK	3.98 c	3.52 b	38.98 c	8.47 ab	0.22 d	0.60 d
YAM	5.36 ab	4.70 a	177.33 a	8.23 c	0.30 b	1.40 b
YCK	4.87 bc	3.68 b	105.82 b	8.54 a	1.39 a	0.90 c

**Table 5 plants-13-02056-t005:** Changes in the structure of rhizosphere soil microbial communities of maize under different treatments.

Microorganisms	Treatment	Kingdom	Phylum	Class	Order	Family	Genus	Species
Bacteria	SAM	1	42	131	313	512	924	1649
	SCK	1	42	134	331	545	997	1769
	YAM	1	42	134	329	545	995	1763
	YCK	1	41	134	326	530	926	1597
Fungi	SAM	1	8	25	59	119	201	271
	SCK	1	7	25	56	106	183	244
	YAM	1	7	25	57	114	187	265
	YCK	1	6	20	41	71	94	116

## Data Availability

The data is contained within the manuscript.

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
