# Peer review of "Effects of AMF on Maize Yield and Soil Microbial Community in Sandy and Saline Soils"

_plants, 2024, doi:10.3390/plants13152056_

Round 1

Reviewer 1 Report

Comments and Suggestions for Authors

Line 12, it is suggested to remove “conditions”.

Lines 27-29 suggest changing to something less emphatic, commenting that the study shows evidence on the benefits of the use of AMF inoculants in corn production (note the type of maïz).

Lines 76-78: it is suggested not to repeat the information in the Table into the text. Also, in the Tables and figures, to describe what the initials mean.

Line 119, section 2.2.2. It is suggested to present first the results of the diversity itself and then, the results of the application of the indices on it!! The quality of Figure 1 is low and the numerals cannot be seen well. It is suggested to repeat what the treatments means for any image taken out of context!!

Line 263. It is requested that it be specified more data about the inoculant comes from the acquired site: provenance, form of presentation, origin, quality control (spores, roots, external mycelium??). If itself, the inoculant comes with bacteria, if yes, is already a change for the communities not precisely linked to what they relate in the study.

Line 271: it is suggested to put the measure in SI of “mu”.

Line 272: it is not clear what was meant by this. Specify.

Line 316: according to the data presented in all the Tables, the analysis does not correspond to a one-way ANOVA since soils versus inoculants are compared. It is not understood why the results have statistics with a Post Hoc if there are no variances , error or deviation! Some variables must have been triplicated like the roots but for the soil this is not seen. It is a single data from a single composite sample??

Comments on the Quality of English Language

It is understandable and readable, only some sentences require consideration.

Author Response

Comments 1:  Line 12, it is suggested to remove “conditions”.

Response 1: Thank you for your careful review of our manuscript and your valuable comments.  Based on your suggestion, we have deleted the "condition" in line 12 of the manuscript to improve the flow of the manuscript.

Comments 2: Lines 27-29 suggest changing to something less emphatic, commenting that the study shows evidence on the benefits of the use of AMF inoculants in corn production (note the type of maize).

Response 2:  Thank you for your suggestion regarding lines 27-29 of the manuscript. We have revised the statement to provide a more nuanced and evidence-based description of the study findings related to the benefits observed from using AMF inoculants in spring maize production. Modified: "This study demonstrated that the use of AMF as a bio-fungal fertilizer was effective in improving spring maize yields, especially in terms of yield increase and quality stability in sandy and saline soils, thereby contributing to safe and sustainable cropping practices." We believe this revision better reflects the study's outcomes and conclusions while maintaining clarity and accuracy.

Comments 3: Lines 76-78: it is suggested not to repeat the information in the Table into the text. Also, in the Tables and figures, to describe what the initials mean.

Response 3: Thank you for your valuable comments and suggestions. We have made the following changes based on your comments. We have removed the duplication of table information in lines 76-78 to avoid repetition. In addition, we have added explanations for the acronyms used in the analysis of results in lines 76-79, which we believe will improve the readability of the text. We thank you for your thorough review and believe these changes improve the clarity and quality of the manuscript.

Comments 4: Line 119, section 2.2.2. It is suggested to present first the results of the diversity itself and then, the results of the application of the indices on it!! The quality of Figure 1 is low and the numerals cannot be seen well. It is suggested to repeat what the treatments means for any image taken out of context!!

Response 4: Thank you for your thorough review of our manuscript and your valuable suggestions. We reviewed the manuscript once again. Please allow me to explain our analysis approach. We first listed the indices methods used to analyze Alpha diversity, making it easier for readers to understand our analysis approach clearly. Then, we analyzed the changes in diversity under saline-alkaline and sandy soil conditions with inoculation compared to CK. Using the analysis of soil microbial diversity under saline-alkaline conditions as an example, we introduced the higher microbial diversity in the AMF treatment under saline-alkaline conditions. Following this, we provided a detailed analysis of the diversity results for each index. We believe this allows for a smooth and clear reading experience for the readers. Regarding the issue you mentioned about the low quality of Figure 1, we redrew the figure and enlarged it as much as possible to help you better understand the details. We also added detailed explanations for each figure in the captions as you suggested. We hope these revisions will make your reading experience more fluent. Thank you once again for your valuable feedback.

Comments 5: Line 263. It is requested that it be specified more data about the inoculant comes from the acquired site: provenance, form of presentation, origin, quality control (spores, roots, external mycelium??). If itself, the inoculant comes with bacteria, if yes, is already a change for the communities not precisely linked to what they relate in the study.

Response 5: Thank you for your valuable suggestions regarding the details of the AMF inoculant. The data on AMF inoculants has now been included in line 267 of the manuscript. Revision: "The initial host of the AMF inoculant used in this study was saffron, produced in Dangxiong, Tibet. It was subsequently expanded with white clover and corn. The inoculum was a mixture of rhizomes and soil with a spore count of 13 spores/gram. The inoculum itself does not contain any bacteria. We believe this additional information addresses your concerns and provides the necessary details about the AMF inoculant used in our study. Thank you for your constructive comments, which helped us to improve the manuscript.

Comments 6: Line 271: it is suggested to put the measure in SI of “mu”.

Response 6: Thank you for your careful review of the manuscript. We have revised the text to include the measure of "mu" in SI units for clarity. The revised text now reads: "Among them, the SAM and YAM treatments received 1 kg of fungal fertilizer per 666.67 m2 (1 mu) at the corn pulling stage, and each plot area was 45 m² (4.5 m × 10 m)." Thanks again for your valuable suggestions.

Comments 7: Line 272: it is not clear what was meant by this. Specify.

Response 7: Thank you for your question. We have revised the sentence in columns 274-276 of the manuscript to specify what is meant by "other management measures" for clarity. The revised text now reads: "Other management measures, such as irrigation, weed control, and pest management, were kept consistent with standard field management practices to ensure uniformity across all treatments." Thank you again for your valuable suggestions.

Comments 8: Line 316: according to the data presented in all the Tables, the analysis does not correspond to a one-way ANOVA since soils versus inoculants are compared. It is not understood why the results have statistics with a Post Hoc if there are no variances, error or deviation! Some variables must have been triplicated like the roots but for the soil this is not seen. It is a single data from a single composite sample??

Response 8: Thank you for your insightful feedback. We understand your concern about the statistical analysis in the manuscript. We assume that you read the article and thought that there were two trial factors, one for soil conditions and one for AMF application. In fact the factors we studied were consistent with your understanding. However, when it comes to the analysis of specific root conformation, soil physical and chemical properties, and other indicators, we combined the two factors of soil and AMF into one treatment and looked at them as a whole. This changed our study to four treatments, SAM, SCK, YAM and YCK, and we replicated each treatment three times so that the variance could be calculated and analyzed. We hope this explanation clarifies the rationale and methodology behind our statistical analysis. Thank you again for your detailed and constructive feedback.

Reviewer 2 Report

Comments and Suggestions for Authors

The manuscript is scientifically interesting and offers a punctual evaluation of the incidence of arbuscular mycorrhizal fungi on maize growth and production in different soil conditions.

The Introduction gives a rapid overview of the state of the art. Results are properly expressed and discussed. Materials and method are clear, such as the Conclusions.

Nevertheless, here below some punctual comments:

-The title is not completely clear and exhaustive. I kindly invite the authors to re-think to it.

-Introduction:

--lines 34-37: factors negatively affecting the reduction in productivity could be extended beingg the result of many factors.

--lines 49-50: the concept is too generic in the present form.

Results:

--line 73: what is SAM? Please spell here being only explained in the abstract but not in the manuscript, up to this line.

--line 74: what are SCK and YAM? As before.

--line 76: what is YCK? As before.

--line 85 and followings: many people without interactions with China does not know how big is a "mu". Please indicate a conversion rate or express it [also] in other international unit/s.

--why in table 2 the datas are expressed per hectare and not mu?

--please put all the names of microorganisms in italicus.

Materials and methods:

--limited attention have been posed on maize characteristics and cultivation practice. Do the authors think that the type of plants (size and hardness of the root system for instance) have a secondary importance in respect to the other elements considered? Otherwise, please give more details.

Comments on the Quality of English Language

No extra comments

Author Response

Comments 1: The manuscript is scientifically interesting and offers a punctual evaluation of the incidence of arbuscular mycorrhizal fungi on maize growth and production in different soil conditions. The Introduction gives a rapid overview of the state of the art. Results are properly expressed and discussed. Materials and method are clear, such as the Conclusions.

Response 1: Thank you for your positive and constructive feedback on our manuscript. We are delighted to hear that you found our study scientifically interesting and appreciated our evaluation of the incidence of arbuscular mycorrhizal fungi on maize growth and production under different soil conditions. We appreciate your comments on the Introduction, Results, Discussion, Materials and Methods, and Conclusions sections. Your recognition of the clarity and thoroughness of these sections is highly valued. We have carefully considered your feedback and are committed to making any necessary improvements. Should there be any specific suggestions or further recommendations you have in mind to enhance our manuscript, we are more than willing to address them in a revised version.

Comments 2: The title is not completely clear and exhaustive. I kindly invite the authors to re-think to it.

Response 2:  Thank you for your valuable feedback. We understand the importance of having a clear and comprehensive title that accurately reflects the content and scope of the study. Based on your suggestion, we have reconsidered the title to better capture the essence of our research. The revised title is “Effect of AMF on maize yield and soil microbial community in sandy and saline soils” We believe this revised title clearly communicates the main focus of our study, highlighting both the application of AMF and the specific soil conditions investigated.

Comments 3: Introduction: lines 34-37: factors negatively affecting the reduction in productivity could be extended being the result of many factors.

Response 3:  Thank you for your insightful feedback on our manuscript. You suggested expanding the factors that negatively affect productivity in lines 34-37 of the introduction. We have revised this section. We hope that the revised introduction meets your expectations and thank you again for your valuable comments.

Comments 4: lines 49-50: the concept is too generic in the present form.

Response 4: Thank you for your suggestion. We have revised the sentence to include specific examples that illustrate how AMF interactions can influence soil microbial communities. Revised Sentence: “AMF can interact with various soil microorganisms, such as rhizosphere bacteria like Bacillus and Pseudomonas, and soil fungi including Aspergillus and Trichoderma, leading to significant changes in the composition, diversity, and functionality of the soil microbial community.” These examples provide specific types of microorganisms that are affected by AMF interactions, thereby illustrating the potential breadth of influence on soil microbial dynamics.

Comments 5: Results: line 73: what is SAM? Please spell here being only explained in the abstract but not in the manuscript, up to this line.

Response 5: Thank you for your valuable comments on our manuscript. We appreciate your suggestion to spell out an explanation of the word "SAM" in the manuscript, as the manuscript only explains the word "SAM" in the abstract and not in the main text. We have revised the manuscript by adding the explanation of SAM in line 76 of the manuscript to make it clearer. We hope this revision will make the manuscript clearer. Thank you again for your valuable comments.

Comments 6: line 74: what are SCK and YAM? As before.

Response 6: Thank you for your valuable feedback on our manuscript. We appreciate your suggestion to spell out an explanation of "SCK and YAM" in the manuscript, as the manuscript only explains "SCK and YAM" in the abstract and not in the main text. We have revised the manuscript by adding the explanation of SCK and YAM in lines 77 and 78 of the manuscript to make it clearer. We hope this revision will make the manuscript clearer. Thank you again for your valuable comments.

Comments 7: line 76: what is YCK? As before.

Response 7: Thank you for your valuable suggestions for our study. We appreciate your suggestion to spell out an explanation of the word "YCK" in the manuscript, as the manuscript only explains the word "YCK" in the abstract and not in the main text. We have revised the manuscript by adding the explanation of YCK in line 79 of the manuscript to make it clearer. We hope this revision will make the manuscript clearer. Thank you again for your valuable comments.

Comments 8: line 85 and followings: many people without interactions with China does not know how big is a "mu". Please indicate a conversion rate or express it [also] in other international unit/s.

Response 8: Thank you for your careful review of the manuscript. We have revised the text to include the measure of "mu" in square meters and provided a conversion rate for clarity. The revised text now reads: "As shown in Table 2, the application of AMF significantly increased the effective number of ears and the number of grains per ear, thereby effectively enhancing the corn yield. The corn yield of SAM was 811.5 kg/666.67 m² (1 mu = 666.67 square meters), representing increases of 25.77%, 7.56%, and 10.56% compared to SCK in terms of yield, effective ear number, and grain number per ear, respectively. The corn yield of YAM was 681.4 kg/666.67 m², with increases of 18.63%, 3.72%, and 11.22% compared to YCK in terms of yield, effective ear number, and grain number per ear, respectively." Thank you again for your valuable comments.

Comments 9: why in table 2 the datas are expressed per hectare and not mu?

Response 9: Thank you for your valuable feedback. We have revised the units in Table 2 to be consistent with the rest of the manuscript. The data are now expressed per 666.67 m² (1 mu = 666.67 square meters) to align with the measurements used throughout the study. This change ensures clarity and consistency for all readers, including those who may not be familiar with the "mu" unit. Thank you again for your suggestion.

Comments 10: please put all the names of microorganisms in italicus.

Response 10: Thank you for your valuable comments on our manuscript. You suggested that the names of all microorganisms be italicized, and we have revised the manuscript accordingly. We hope that this revision will make the manuscript clearer. Thank you again for your valuable comments.

Comments 11: Materials and methods: limited attention have been posed on maize characteristics and cultivation practice. Do the authors think that the type of plants (size and hardness of the root system for instance) have a secondary importance in respect to the other elements considered? Otherwise, please give more details.

Response 11: We appreciate your valuable comments on our manuscript. We appreciate your observations on maize characteristics and growing methods. We agree that these factors are critical and can have a major impact on our findings. In fact, these factors play an important role along with other factors such as soil conditions and microbial interactions. However, in this study, we chose the main corn cultivar Kehe 699 in our experimental site as a single test material and applied the same cultivation practices to each treatment, thus controlling the variables and studying only the effects of AMF on corn and soil microbes. Your suggestion has provided us with new ideas for our subsequent research, and we will subsequently launch a study on the interaction of different maize materials and cultivation methods with maize growth and soil microorganisms. And based on this, we will explore the effects of all three on maize growth and soil microorganisms in an integrated manner. Thank you for emphasizing this important aspect, and we are excited to follow up on this aspect of our research.

Round 2

Reviewer 2 Report

Comments and Suggestions for Authors

Dear authors, thanks for your punctual revisions and improvements.

I just have a few short/simple comments (I am citing the numeration used by the authors in the answer):

Comment 2: the authors reported the new title of the manuscript while in the pdf of the manuscript the title was not changed.

Comment 9: the authors clearly expressed the area considered and recalculated the results. But why the equivalence is reported both at line 89 (with an error in the spelling) and at line 276? In Table 2, because the equivalence is reported again at the bottom of the same table, maybe it is possible to write "1 mu" in the headspace instead of 666.67 m2, saving space (esthetical presentation).

Comment 10: not all the names have been corrected. 

Author Response

Response to Reviewer 2 Comments

Comments 2:  the authors reported the new title of the manuscript while in the pdf of the manuscript the title was not changed.

Response 2:  Thank you for your approval of our revision. The revised title was changed in our revised manuscript marked in red.

Comments 9:  the authors clearly expressed the area considered and recalculated the results. But why the equivalence is reported both at line 89 (with an error in the spelling) and at line 276? In Table 2, because the equivalence is reported again at the bottom of the same table, maybe it is possible to write "1 mu" in the headspace instead of 666.67 m2, saving space (esthetical presentation).

Response 9: Thank you for your suggestion. The same equivalence in Line 276 was deleted and the headspace in Table 2 was also corrected.

Comments 10: please put all the names of microorganisms in italicus.

Response 10: Sorry for my carelessness. I read the whole manuscript and italicized all the names of all microorganisms. Thank you again for your valuable comments.